# Prenatal Detection of Rapid Progressive Changes in Massive Lymphangioma from Flank to the Lower Extremity

**DOI:** 10.3390/diagnostics13132130

**Published:** 2023-06-21

**Authors:** Saipin Pongsatha, Phudit Jatavan, Panisa Hantrakun, Theera Tongsong

**Affiliations:** Department of Obstetrics and Gynecology, Faculty of Medicine, Chiang Mai University, Chiang Mai 50200, Thailand; saipin.pongsatha@cmu.ac.th (S.P.);

**Keywords:** fetus, lymphangioma, prenatal diagnosis, lower extremity, ultrasound

## Abstract

Lymphangioma is a congenital anomaly in which abnormal lymphatic drainages localize to form a benign mass, but it has the tendency to grow in size and the potential to infiltrate surrounding structures, causing devastating effects and leading to severe morbidity. The most common site of lymphangioma is the neck region (cystic hygroma colli), whereas lymphangioma in the lower limbs is very rare, accounting for only 2% of cases. Accordingly, the prenatal diagnosis of lymphangioma of the lower limbs has been scarcely reported. This study describes two cases of lymphangioma of the lower limbs, focusing on unique sonographic features and the natural course of rapidly progressive changes, which is different from nuchal lymphangioma. Based on previous isolated case reports together with our two cases, lymphangioma of the lower limbs usually develops in the second trimester, tends to have rapidly progressive changes, and is unlikely to be associated with aneuploidy and structural anomalies. Diagnoses can be made by using sonographic findings pertaining to the subcutaneous complex and multi-septate anechoic cystic lesions in the lower limbs, the latter of which can infiltrate visceral structures. Prenatal detection can be helpful in laying the groundwork for providing counseling to the parents and the planning of management strategies, i.e., opting to terminate the pregnancy, revising delivery plans, and looking towards the postnatal management of the infant.

## 1. Introduction

Lymphangioma is a congenital anomaly in which abnormal lymphatic drainages localize to form a benign mass. The abnormality may develop as early as the 6th week of gestation. Because of a developmental defect, the lymphatic system cannot communicate properly with the venous system, leading to dilatation of the afferent lymphatic channels and culminating in an internal buildup of fluid and cyst formation. Though lymphangioma is pathologically benign, it has a tendency to rapidly grow in size and the potential to infiltrate surrounding structures, causing devastating effects and leading to severe morbidity. Lymphangioma can occur in a variety of anatomical locations. The most common area is the neck region (75%), known as cystic hygroma colli or nuchal lymphangioma, followed by the axillary region (20%), retro-peritoneum and abdominal viscera (2%), limbs and bones (2%), and cervico-mediastinum (1%) [1]. To the best of our knowledge, prenatal diagnosis of lymphangioma of the lower limbs has scarcely been reported [2,3,4,5]. This study describes two cases of lymphangioma of the lower limbs, focusing on unique sonographic features and the natural course of rapidly progressive changes, which differ from nuchal lymphangioma.


**Case 1:**


A 36-year-old primigravid woman naturally conceived and attended our antenatal care clinic as a high-risk pregnancy because of elderly gravida. Her prenatal care course was unremarkable. She had no significant familial diseases or underlying medical diseases. Nevertheless, amniocentesis for chromosome study at 16 weeks of gestation was performed because of elderly gravida. The fetus had a normal karyotype; 46, XX. An ultrasound examination before amniocentesis revealed normal structures, including both lower limbs (Figure 1A). An ultrasound examination for fetal anomaly screening at 20 weeks of gestation showed normal fetal biometry, normal amniotic fluid volume, and normal placenta. Upon surveying the fetus’ anatomy, the ultrasound showed multi-septate cystic mass at the right lower flank and right internal pelvic region, extending down to the buttock, thigh, and lower leg. The right flank, perineum, buttock, thigh, and lower leg exhibited extensive edema with subcutaneous anechoic multi-septate cysts involving the entire right lower extremity, as presented in Figure 1B–F. Massive lymphangioma of the right lower abdomen and right lower extremity was diagnosed based on the findings from the ultrasound. Aspiration of the fluid from the cystic space of the right flank was performed. Cytology revealed abundant lymphocytes and macrophages, confirming the diagnosis of lymphangioma. The rapid progressive change in the lesion, which involved the entirety of the lower limb, perineum, and intraabdominal pelvic regions, severe morbidity was determined; hence, the couple opted to have the pregnancy terminated. A postnatal examination confirmed the prenatal findings.


**Case 2:**


A 34-year-old primigravid woman had an uneventful antenatal course of pregnancy. The pregnancy was naturally conceived and her prenatal care course was unremarkable. She had no significant familial diseases or underlying medical diseases. Non-invasive prenatal testing (cell-free fetal DNA test) revealed a low risk of aneuploidy. An ultrasound examination for fetal anomaly screening at 20 weeks of gestation showed normal fetal biometry, normal amniotic fluid volume, and normal placenta. Nevertheless, upon surveying the fetus’ anatomy, mild edema of the fetal right thigh was noted (Figure 2A), while the other anatomical structures were unremarkable, except for the abnormal shape of the left first toe. On the follow-up ultrasound examination at 24 weeks of gestation, the right flank connecting to the right pelvic region and extending down to the right thigh became extensively edematous, as evidenced by subcutaneous complex, multi-septate, avascular, and anechoic cysts, as presented in Figure 2B–F. Based on the findings from the ultrasound, lymphangioma of the right thigh and flank was diagnosed. Because of rapid growth and severe morbidity, the couple decided to terminate the pregnancy. A postnatal examination and autopsy confirmed the prenatal findings, as presented in Figure 2G,H.

## 2. Discussion

Lymphangioma is a congenital anomaly in which abnormal lymphatic drainages localize to form a benign mass. Despite being pathologically benign, lymphangioma has a tendency to rapidly grow in size and the potential to infiltrate surrounding structures, causing devastating effects and leading to severe morbidity. The most common site of lymphangioma is the neck region (nuchal lymphangioma or cystic hygroma colli), whereas lymphangioma of the lower limbs is very rare, accounting for only 2% of cases. The prenatal diagnosis of lymphangioma of the lower limbs has scarcely been reported. This study describes two cases of lymphangioma of the lower limbs, focusing on unique sonographic features and the natural course of rapidly progressive changes. According to this study and previous case reports, the diagnosis of lymphangioma of the lower limbs can be made on the basis of sonographic findings of the avascular subcutaneous complex and multi-septate anechoic cystic lesions in the lower limbs, flank, or pelvis, which can infiltrate visceral structures. The differential diagnosis includes teratoma, isolated hemangiomas, Klippel–Trenaunay–Weber syndrome, Wilms’ tumor, and neuroblastoma.

Interestingly, aneuploidy is very common among fetuses with nuchal lymphangioma, with a rate of approximately 70% of affected fetuses, the majority being 45XO [6]. Nevertheless, to the best our knowledge, non-nuchal lymphangiomas are probably not associated with aneuploidy, as seen in our two cases and other reports [1,3,7,8,9,10,11]. However, because of the paucity of available data, it is uncertain whether or not lymphangiomas of the lower extremities carry a significant risk for aneuploidy, and chromosomal studies may be individualized. Likewise, nuchal lymphangiomas are also commonly associated with other malformations or syndromes; however, non-nuchal lymphangiomas are likely isolated and not associated with other structural anomalies. Additionally, nuchal lymphangiomas are commonly detected in the first trimester or early second trimester, while lymphangiomas of the lower extremities tend to be late-occurring malformations likely to go undetected during anomaly screening via ultrasound at mid-pregnancy. Notably, isolated nuchal lymphangiomas associated with a normal karyotype may spontaneously regress with advancing gestational age, whereas the course of disease in our cases was rapidly progressive with gestational age. The findings of rarity of aneuploidy, isolated abnormality, late-occurring anomaly, and rapid progression suggest that lymphangiomas of the lower extremities have a different natural course compared to nuchal lymphangiomas. Nevertheless, more contributions need to be made to the literature to confirm our observations.

The overall prognosis of fetal lymphangiomas is poor, with a mortality rate of 50 to 100%. However, little is known about the prognosis of non-lymphangiomas. Most cases in several reports had poor prognosis, though some had good outcomes [7]. Therefore, a decision may be made regarding the termination of the pregnancy or the decision may be individualized based on the severity or progressive changes detected upon follow-up scans. Accordingly, prenatal detection can be helpful in laying the groundwork for providing counseling to the parents and the planning of management strategies, i.e., opting to terminate the pregnancy, revising delivery plans, and looking towards the postnatal management of the infant.

## 3. Conclusions

This study, together with previous cases reports, illustrates the interesting points of lymphangiomas of the fetal lower extremities as follows: (1) The diagnosis of lymphangioma should be suspected whenever an avascular, complex cystic subcutaneous mass with septations is noted in a fetus. (2) Fetal non-nuchal lymphangioma is unlikely to be associated with aneuploidy and structural anomalies as these differ from nuchal lymphangioma. (3) The extent of growth can affect adjacent structures in the fetal pelvis as well as the perineum and may be associated with poor prognosis. (4) Serial sonographic examinations are important because the early detection of rapid growth of the mass is helpful in laying the groundwork for providing counseling to the parents and the planning of management strategies, i.e., opting to terminate the pregnancy, revising delivery plans, and looking towards the postnatal management of the infant. (5) The first sonographic sign in our two cases were asymmetry of both thighs due to edema of the affected side.

## Figures and Tables

**Figure 1 diagnostics-13-02130-f001:**
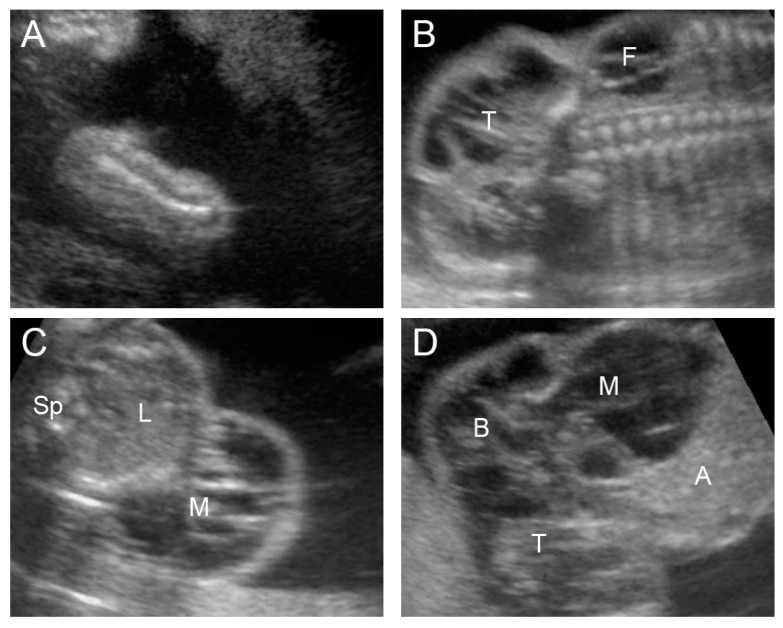
Ultrasound findings of the fetus (case 1) at 16 weeks (**A**) and 20 weeks (**B**–**F**) of gestation; (**A**) normal right lower limb at 16 weeks of gestation; (**B**) coronal scan of lower abdomen shows multi-septate cystic mass at the right flank (F), extending down to the right thigh (T). (**C**) Cross-sectional scan of lower abdomen shows multi-septate cystic mass at the right flank (M) (Sp: spine; L: liver). (**D**) Sagittal scan of lower abdomen (A) shows multi-septate cystic mass in the pelvis (M), connecting to the buttock (B) and thigh (T). (**E**) Longitudinal view of the right thigh shows markedly edematous perineum (P), thigh (T), and knee joint (K), as well as the lower leg. (**F**) Plantar view of the right foot shows obvious edema.

**Figure 2 diagnostics-13-02130-f002:**
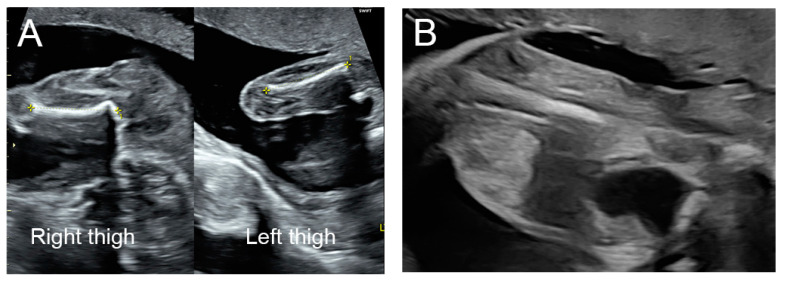
Ultrasound findings of the fetus (case 2) at 20 weeks (**A**) and 24 weeks (**B**–**F**) of gestation; (**A**) Ultrasound screening for fetal anomaly at 20 weeks of gestation, mild edema of the fetal right thigh is noted. Long-axis view of the right thigh (left side) and the left thigh (right side) shows discordant in size. (**B**) Long-axis view of the right thigh (T) shows multi-cystic anechoic lesions with marked edema through the thigh and the edema, ending sharply above the right knee (K). (**C**) Coronal scan with color slow-flow Doppler ultrasound of the lower abdomen including upper right thigh shows complex multi-cystic anechoic cyst without vascular flow at the right flank (F) as well as pelvic region adjacent to the bladder (B) and upper thigh (T). (**D**) Cross-sectional scan of the abdomen shows multi-septate cystic lesions at the right flank extending up to the upper abdomen. (**E**) Cross-sectional scan of the lower abdomen shows complex multi-septate anechoic cystic lesions at the right flank, pelvic region adjacent to the bladder (B), and the right thigh (T). (**F**) Cross-sectional scan of the fetal chest at the level of four-chamber view shows subcutaneous anechoic fluid collection (*) surrounding the rib. (**G**) Gross finding of the lower extremities of the neonate, comparing the enlarged right thigh with the normal left thigh. (**H**) Gross findings of the opened pelvis and right thigh showing lymphatic cysts infiltrating the inner pelvis, the right flank, and the right thigh.

## Data Availability

The data of this report are available from the corresponding authors upon request.

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
