# Peer review of "Prenatal Detection of Rapid Progressive Changes in Massive Lymphangioma from Flank to the Lower Extremity"

_diagnostics, 2023, doi:10.3390/diagnostics13132130_

Round 1
Reviewer 1 Report
A relevant contribute to the knowledge of the natural history of this condition, well presented and illustrated. I mostly appreciated the syntetic esposition.
correctly written
Author Response
Comments and Suggestions for Authors
A relevant contribute to the knowledge of the natural history of this condition, well presented and illustrated. I mostly appreciated the syntetic esposition.
Comments on the Quality of English Language
correctly written
Response: Thank you.
Reviewer 2 Report
The topic suggestion change "Prenatal detection of rapid progressive change of massive lymphangioma of the lower extremity" to "Prenatal detection of rapid progressive change of massive lymphangioma from flank to the lower extremity" . Because both of cases massive lymphangioma include flank to buttock and lower extremity and not only restrict the lower extremity.

Author Response
Comments and Suggestions for Authors
The topic suggestion change "Prenatal detection of rapid progressive change of massive lymphangioma of the lower extremity" to "Prenatal detection of rapid progressive change of massive lymphangioma from flank to the lower extremity". Because both of cases massive lymphangioma include flank to buttock and lower extremity and not only restrict the lower extremity.
Response: The title has been changed as suggested.
Reviewer 3 Report
The authors describe two cases of rapid progressive changes of massive lymphangioma of the lower extremity.
In both cases, the authors have presented attractive images that reflect the situation well.
The cases are described in detail and all important points are briefly discussed (genetics, follow-up, prognosis and also the possibility of terminating the pregnancy).
The clinical picture described is not extremely rare, but it can take on massive forms, where a decision must be made about further action (continuation of the pregnancy or not)
The paper is a good reference for clinicians.
The paper is well written. minor englisch correction are needed throughout the text.
Author Response
Comments and Suggestions for Authors
The authors describe two cases of rapid progressive changes of massive lymphangioma of the lower extremity.
In both cases, the authors have presented attractive images that reflect the situation well.
The cases are described in detail and all important points are briefly discussed (genetics, follow-up, prognosis and also the possibility of terminating the pregnancy).
The clinical picture described is not extremely rare, but it can take on massive forms, where a decision must be made about further action (continuation of the pregnancy or not)
The paper is a good reference for clinicians.
Response: Thank you for the comment.
Reviewer 4 Report
I want to thank the Multidisciplinary Digital Publishing Institute and Ms. Thalia Pang, assistant editor, for the opportunity to review this article.
I have read the article by Pongsatha et al. entitled “Prenatal detection of rapid progressive change of massive lymphangioma of the lower extremity”.
The authors present two cases of lower extremity lymphangioma.
Overall, the manuscript is interesting, read well, and adequately written. Nevertheless, there are essential concerns regarding this paper.
General:
In this study, the ultrasonographic findings are quite spectacular and much-needed data needing reporting and investigating.
Please make sure to correct the minor spelling errors throughout the manuscript.
After the introduction I suggest the authors that describe each case as follows:
Each case shall be described in the most detailed as possible.
Case 1: A 36-year-old primigravid……” complete the description of the case. When you describe the ultrasound findings then indicate Fig.1A and so on.
In the figure you should include the following: Figure 1. Ultrasound findings of the fetus diagnosed at mid-trimester screening scan. Fig A: “normal right lower limb at 16 weeks”. The description of this figure from Fig B to Fig F is adequate.
Case 2: please correct it as case 1.
Please perform a proper discussion and conclusion for this paper

Author Response
Comments and Suggestions for Authors
I want to thank the Multidisciplinary Digital Publishing Institute and Ms. Thalia Pang, assistant editor, for the opportunity to review this article.
I have read the article by Pongsatha et al. entitled “Prenatal detection of rapid progressive change of massive lymphangioma of the lower extremity”.
The authors present two cases of lower extremity lymphangioma.
Overall, the manuscript is interesting, read well, and adequately written. Nevertheless, there are essential concerns regarding this paper.
General:
In this study, the ultrasonographic findings are quite spectacular and much-needed data needing reporting and investigating.
Please make sure to correct the minor spelling errors throughout the manuscript.
After the introduction I suggest the authors that describe each case as follows:
Each case shall be described in the most detailed as possible.
Case 1: A 36-year-old primigravid……” complete the description of the case. When you describe the ultrasound findings then indicate Fig.1A and so on.
In the figure you should include the following: Figure 1. Ultrasound findings of the fetus diagnosed at mid-trimester screening scan. Fig A: “normal right lower limb at 16 weeks”. The description of this figure from Fig B to Fig F is adequate.
Case 2: please correct it as case 1.
Please perform a proper discussion and conclusion for this paper
Response: In the revised MS, each case is firstly described in details and followed by Figure as suggested. We have rearranged the sequence of presentation and add some more details of the case description, as presented in red.
Round 2
Reviewer 4 Report
I want to thank the Multidisciplinary Digital Publishing Institute for the opportunity to review this article.
I have read the article by Pongsatha et al. entitled “Prenatal detection of rapid progressive change of massive lymphangioma from flank to the lower extremity.”
The authors present two cases of lower extremity lymphangioma.
Overall, the manuscript is fascinating, read well, and adequately written.
Nevertheless, there is still one essential concern regarding this paper.
This paper is still missing a proper discussion and conclusion.

Author Response
Reviewer 4
Comments and Suggestions for Authors
I want to thank the Multidisciplinary Digital Publishing Institute for the opportunity to review this article.
I have read the article by Pongsatha et al. entitled “Prenatal detection of rapid progressive change of massive lymphangioma from flank to the lower extremity.”
The authors present two cases of lower extremity lymphangioma.
Overall, the manuscript is fascinating, read well, and adequately written.
Nevertheless, there is still one essential concern regarding this paper.
This paper is still missing a proper discussion and conclusion.
Response: In revised MS, discussion and conclusion are added as suggested, as highlighted in red at the end of MS. Nevertheless, according to the “author instruction for “interesting image”, no regular manuscript text introduction/methods results/discussion should be included. Therefore, we add discussion and conclusion in a succinct format.
Round 3
Reviewer 4 Report
Well done !
Congratulations.